# Safety and Immunogenicity of Trivalent Oral Polio Vaccine in Vaccinated Children and Vaccine-Naïve Infants: A Phase 4 Study

**DOI:** 10.3390/vaccines12090953

**Published:** 2024-08-23

**Authors:** Luis Rivera Mejía, Lourdes Peña Mendez, Ricardo W. Rüttimann, Chris Gast, Ananda Sankar Bandyopadhyay

**Affiliations:** 1Hospital Universitario Maternidad Nuestra Señora de la Altagracia, Fundación Dominicana de Perinatología PROBEBE, Calle Pedro Henríquez Ureña #49, Santo Domingo 10205, Dominican Republic; lrivera@probebe.org.do; 2Clínica Cruz Jiminian, Av Ortega y Gasset 90, Santo Domingo 10501, Dominican Republic; lourdespena@medyvacinternacional.com; 3Fighting Infectious Diseases in Emerging Countries (FIDEC), 2050 Coral Way, Suite 407, Miami, FL 33145, USA; 4Independent Biostatistician Consultant, Seattle, WA 98029, USA; 5Bill and Melinda Gates Foundation, Seattle, WA 98109, USA; ananda.bandyopadhyay@gatesfoundation.org

**Keywords:** poliomyelitis, eradication, trivalent oral polio vaccine

## Abstract

In the context of polio eradication, novel oral polio vaccines for type 2 (nOPV2) were developed, and types 1 and 3 polioviruses are being developed. We aimed to generate trivalent oral poliovirus vaccine (tOPV) safety and immunogenicity data as a reference for comparing with novel OPV formulations. This was a single-center, open-label, phase 4 study in March 2016 in the Dominican Republic with healthy children previously vaccinated with ≥3 doses of tOPV receiving one dose of tOPV and vaccine-naïve infants receiving 3 doses of tOPV. Safety and immunogenicity were assessed. No serious adverse reactions or important medical reactions were reported. Seroconversion (SC) rates at Day 28 in children were 32.7%, 36.7%, and 46.9% for types 1, 2, and 3, respectively, and seroprotection (SP) rates 28 days after one dose increased from 89.8% at baseline to 93.9%, 98.0% to 100%, and 83.7% to 98.0% for types 1, 2, and 3, respectively. In infants, SC rates were 88.5%, 98.1%, and 96.2% for types 1, 2, and 3, respectively. SP rates at Day 84 were 93.3%, 100%, and 96.2% for types 1, 2, and 3, respectively. This information can be used as a reference to compare with novel monovalent or trivalent OPVs under development.

## 1. Introduction

The current epidemiology of polioviruses and the available prevention tools have led the global eradication program to explore innovative options for the final stretch to achieve and sustain the eradication of all forms of polioviruses. The Global Polio Eradication Initiative (GPEI) started its fight against polio in 1988, achieving a significant decline in polio cases of more than 99% over the subsequent three and a half decades [1]. Significant progress towards the eradication of polio was made in the last ten years; for instance, the World Health Organization (WHO) Southeast Asia Region was declared free of poliovirus in 2014; then, in 2015 wild poliovirus type 2 (WPV2) was declared eradicated and so was WPV3 in 2019; and the most recent region to be certified free of WPV by the WHO was the African Region, in August 2020 [2,3,4].

Nevertheless, significant challenges remain in the last mile to stop all forms of polioviruses; thus, the GPEI has made an urgent request to governments, social groups, and all other interested parties regarding the responsibility and efforts to make the eradication a reality [2,5]. The actions proposed via the Polio Eradication Strategy 2022–2026 include two main goals: (1) Stop WPV transmission in endemic countries (Afghanistan and Pakistan), and (2) Stop circulating vaccine-derived poliovirus (cVDPV) transmission and prevent outbreaks in non-endemic countries, given that cVDPV type 2 (cVDPV2) outbreaks have been detected in four of the six WHO geographical regions [2].

The successful results on the eradication of WPV2 and 3 are primarily attributable to global routine immunizations with the Sabin trivalent oral poliovirus vaccine (tOPV), which has the advantage of providing mucosal immunity and, therefore, is effective at interrupting poliovirus transmission and also is low cost and easy to administer. Since it was licensed in the United States in 1962, tOPV, which incorporates poliovirus types 1, 2, and 3, was the preferred vaccine for polio control and eradication. Global use of this vaccine has enabled the eradication of WPV2 and 3 [6,7]. In May of 2016, considering the certification of eradication of WPV2, the use of Sabin OPV2 was stopped for routine and supplementary immunization activities, and a global switch from tOPV to bivalent OPV (bOPV), with Sabin strain types 1 and 3, took place [8]. The use of OPV2 is no longer allowable as per global containment recommendations, and its use is restricted only for responding to cVDPV2 outbreaks in some countries [9]. However, this led to a high risk of cVDPV2 outbreaks in populations with low immunization levels because of the incompletely interrupted transmission of the Sabin strain poliovirus, which can lose its attenuations over time and revert to neurovirulent strains, giving rise to paralytic outbreaks [10,11,12]. Following the epidemiological reports and advances in vaccine development, the WHO Emergency Use Listing (EUL) procedure authorized in November 2020 the use of the now WHO-prequalified type 2 novel OPV (nOPV2), a vaccine modified to be more genetically stable than the Sabin strain to more sustainably interrupt cVDPV2 outbreaks [13,14]. Furthermore, novel vaccine candidates for types 1 and 3 polioviruses and novel trivalent OPV are being developed by incorporating elements from nOPV2, aiming for safer and similar effective vaccines [15].

As the development of novel trivalent OPV and novel monovalent OPVs continues, historical data on Sabin trivalent OPV will be critically important as, following the global cessation of the use of Sabin type 2-containing OPVs, including tOPV, this vaccine can no longer be used in clinical trial settings. Therefore, the only reference point for the future novel trivalent OPV formulation will be the Sabin trivalent OPV data reported here, for contextualization, especially for age-matched infant cohorts. This historical control approach was used in the development of the nOPV2, where data from studies performed before cessation with Sabin monovalent OPV2 were compared to the subsequent data from the studies where nOPV2 was assessed, providing safety and immunogenicity data that ultimately supported the deployment of nOPV2 [16,17]. 

This study was conducted to generate data on the safety and immunogenicity of tOPV to provide historical control data for comparing novel oral polio vaccines after the global withdrawal of routine use of all Sabin OPVs. We report the safety and humoral immunogenicity results of Sabin tOPV in healthy polio-vaccinated children aged 1 to 5 years and in vaccine-naïve infants. The study was performed prior to the cessation of routine use of tOPV.

## 2. Materials and Methods

### 2.1. Study Design and Population

We conducted a single-center, open-label, phase 4 study in Santo Domingo, Dominican Republic, between 7 January and 30 March 2016. Eligible participants were healthy children aged 1 to 5 years who had been previously vaccinated with ≥3 doses of tOPV (Group 1, 50 children) and healthy tOPV-naïve infants aged 6 weeks (Group 2, 104 infants). Group 1 received one dose of tOPV, while Group 2 received 3 doses of tOPV administered 28 days apart. Exclusion criteria comprised any known allergy to a vaccine component, immunodeficient conditions, acute or chronic illnesses, and the presence of an individual in the participant’s household who had received OPV in the previous three months. 

### 2.2. Vaccine and Immunization Procedures

The vaccine used in this study was a licensed SABIN tOPV, Opvero™, manufactured by Sanofi Pasteur, Marcy l’Etoile, France. Each dose of vaccine contained at least 6.0 log 50% cell culture infective dose (CCID_50_) of LS c2ab strain of live attenuated poliovirus type 1, 5.0 log CCID_50_ of P712, Ch, 2ab strain of live attenuated poliovirus type 2, and 5.8 log CCID_50_ Leon I2aIb strain of poliovirus type 3. One dose of vaccine (2 drops, 0.1 mL) from commercial batches was administered using a dropper supplied with the multidose container.

Other vaccines were not administered during the whole study period for Group 1. Still, routine infant vaccinations (excluding polio vaccines but including DTPw-HBV-Hib, pneumococcal conjugate, and rotavirus vaccines) were administered at separate vaccination visits following the national immunization schedule (2–4–6 months of age) in Group 2 infants. Influenza vaccines were allowed in both groups according to the National Immunization Recommendation.

### 2.3. Study Assessments

Co-primary objectives of the study were to assess the safety—expressed as the incidence of serious adverse reactions events considered consistent with a causal association to the study vaccine (SARs), severe adverse reactions (ARs), and important medical reactions (IMRs), after one dose of tOPV in children and three doses of tOPV in 6-week-old infants, and immunogenicity (seroprotection (SP) rates for all 3 serotypes) 28 days after three doses of tOPV in infants. 

Secondary objectives included assessment of the incidence of any serious adverse events (SAEs), adverse events (AEs), and important medical events (IMEs) after one dose of tOPV in children and three doses of tOPV in infants; immunogenicity (seroconversion (SC) rates, median and geometric mean antibody titers (GMTs) against all 3 types) 28 days after a third dose of tOPV in infants and following a single dose of tOPV in children. 

### 2.4. Immunogenicity Assessments

Blood samples were drawn at Days 0, 7, and 28 for children and at Days 0 and 84 (28 days after the 3rd vaccine dose) for infants. 

Neutralizing antibodies against polioviruses 1, 2, and 3 were measured in collected sera using the WHO standard microneutralization assay (WHO EPI GEN 93.9). Antibody assays were performed at the US Centers for Disease Control and Prevention (CDC). GMTs of neutralizing antibodies were calculated using the antilog of the arithmetic mean of log_2_ antibody titers. 

As it was calculated in a similar design study performed by our group that developed the nOPV2, antibody titers were expressed as group median log_2_ titers with 95% CIs, proportions with a reciprocal titer of eight or greater (seroprotection rate), and proportions either becoming seroprotected when seronegative at baseline or displaying fourfold or greater increases in titers from baseline to post-vaccination (seroconversion rate). As some participants had high baseline titer, seroconversion was only calculated for those whose baseline titer allowed observation of a fourfold increase without being above the upper limit of quantitation (median log_2_ of 10·5) [16]. In the case of infants with maternal antibodies before vaccination, the calculation of the titer increase considered the natural decay of these antibodies, assuming an exponential decay model with a half-life of 28 days. 

### 2.5. Safety Assessments

Subjects’ parent(s) or guardian(s) were provided with an electronic diary and a paper diary as backup, which included the definitions of mild, moderate, and severe solicited AEs to facilitate the assessments of the level of functional impairment for each experienced AE.

Parents completed electronic diary cards to record solicited AEs for seven days after each vaccine dose and any unsolicited AEs throughout the study. Parents were also contacted by phone call or text message daily from days 0 to 14 and on day 21 after each vaccine dose. 

All AEs occurring during the study were documented, and the Investigator inquired about the occurrence of AEs/SAEs/IMEs at every study visit/contact. 

Clinical laboratory assessments (hematology, chemistry, and coagulation analyses) were performed. The incidence and description of deviations from standard safety clinical laboratory assessments at Days 0, 7, and 28 after the dose of Sabin tOPV in Group 1 were documented, as well as the incidence and description of deviations from standard safety clinical laboratory assessments at Days 0 and 84 (before and after the third doses of Sabin tOPV in Group 2.

A Data and Safety Monitoring Board (DSMB) monitored the benefit–risk and data integrity of the study.

### 2.6. Statistical Analysis

The total vaccinated (TV) population, defined as all subjects who received at least one dose of the study vaccine, was used for safety and demographics analysis. The per-protocol (PP) population, which consisted of all eligible study participants in the TV population who received all the immunizations scheduled for the group to which they were allocated, was used for immunogenicity analysis.

For the safety analysis, a sample size of 50 participants was estimated to provide adequate data; a follow-up of 45 subjects (assuming a 10% non-evaluability/dropout rate) yielded a 90% probability of detecting an AE rate greater than 0% when the actual AE rate was 5%. The type I error rate was chosen to be alpha = 0.05. 

For the immunogenicity endpoint, overall non-inferiority of a potential future test vaccine to this historical control arm could be declared if all serotypes individually achieved non-inferiority with type I error rate alpha = 0.025 and a non-inferiority margin of 10%. This method controlled type I error rate for the overall non-inferiority comparison at level ≤0.025. It was assumed that the SP rate after administering three doses of the current tOPV and a potential future tOPV in this population would be 97% for each serotype. 

A sample size of 94 evaluable participants per arm was required to achieve 90% power to declare the overall non-inferiority of a trivalent test vaccine to this historical control. Therefore, 104 vaccine-naïve infants were enrolled in the three-dose group, allowing for a 10% dropout/non-evaluability rate. 

Descriptive statistics were provided per group for demographics and other initial subject characteristics, including mean, standard deviation (SD), median, maximum, minimum, and range for continuous variables, and count and percentage for categorical variables. Statistical tests and confidence intervals (CIs) were computed using a two-sided 5% significance level. 

At each pre- and post-vaccination time point where neutralizing antibody titers were obtained, SP and SC rates with 95% CI were computed, cumulative rates of SC and SP, the median of log_2_ antibody titers, and GMTs with 95% CI. Plots of the reverse cumulative distribution of antibody titers were generated.

All AEs were summarized by group, occurrence in causal association to vaccination, and overall. Determination of causal association to the study vaccines was assigned using the individual causality assessment algorithm published by WHO in 2013 [18].

### 2.7. Ethics

The study was conducted by the HUMNSA University Hospital (Hospital Universitario de Maternidad Nuestra Señora de la Alta Gracia) Santo Domingo, Dominican Republic, and the study protocol, informed consent form (ICF), materials provided to subjects, and applicable recruiting material were approved by the HUMNSA Ethical Review Board (ERB) on 8 October 2015 and the National Council for Bioethics in Health (Consejo Nacional de Bioética en Salud, CONABIOS) of the Ministry of Health. The informed consent was obtained after fully explaining the study’s nature and possible consequences to the parents and legal guardians of the participants. 

The study was conducted according to the ethical principles of the Declaration of Helsinki, the International Council for Harmonisation (ICH) of Technical Requirements for Pharmaceuticals for Human Use guideline for Good Clinical Practice (GCP), and the applicable regulatory and country-specific requirements regarding research on human subjects.

Clinical Trial registry NCT02580201.

## 3. Results

### 3.1. Study Subjects

We enrolled 50 children aged 1–5 years (Group 1) and 104 infants 6 weeks of age (Group 2). All participants received scheduled vaccinations and completed the study with no early terminations (Figure 1).

Children and infants had similar demographic characteristics by group (Table 1). The mean age was 148.3 weeks in Group 1 and 5.8 weeks in Group 2.

Most participants in Group 1 (90.0%) had received ≥3 tOPV vaccinations before 1 year of age; 56.0% had also received at least one booster vaccination (at 18 months and 4 years). For individuals who had received a vaccination less than 1 year before the study, the mean interval between the last vaccination and study enrollment was 7.7 months.

All participants in Group 2 received 3 doses of tOPV (the meantime interval between the first and second doses was 28.2 days, and between the second and third doses 28,3 days).

#### 3.1.1. Safety Results

No child or infant experienced any SAR or IMR after receiving tOPV. One participant in Group 2 experienced two ARs: vomiting after dose 1 and abnormal crying after dose 3. (See Appendix A).

Solicited AEs were reported in 7 (14%) children and 39 (37.5%) infants (Table 2). In Group 1, there were 9 AEs considered mild or moderate, non-severe, and none causally associated with the study vaccine. In group 2, 87 AEs were reported, 80 were considered mild or moderate, and 7 were severe. A total of 18 AEs in 12 participants were considered causally associated with the study vaccine. The most frequent causally associated AEs were abnormal crying (8 participants, 1 severe), vomiting (5 participants, 1 severe), and irritability (2 participants, 0 severe).

Unsolicited AEs were observed in 2 (4%) individuals in Group 1, and 20 (19.2%) in Group 2. All were reported as mild or moderate and inconsistent with causal association to immunization.

Three infants (2.9%) in Group 2 experienced a total of 4 SAEs during the study (intestinal amoebiasis and urinary tract infection, dengue, and bronchiolitis in 1 participant each). These SAEs were moderate in severity; none were considered causally associated with immunization. 

No clinically relevant abnormalities in hematology or chemical laboratory assessments were observed. Only 1 child in Group 1 had a clinically relevant coagulation value, fibrinogen values above the normal range on Days 0 and 7, and was considered inconsistent with a causal association to immunization.

#### 3.1.2. Immunogenicity Results

The primary objective was SP in infants. In this Group 2, SP rates for type-specific neutralizing antibodies at baseline were 60.6% for types 1 and 2 and 26.0% for type 3. At day 84, they were 93.3%, 100%, and 96.2% for types 1, 2, and 3, respectively, and SC rates were 88.5%, 98.1%, and 96.2% for types 1, 2, and 3, respectively. (See Appendix A).

For Group 1, SP rates increased from 89.8% at baseline to 93.9% for type 1, 98.0% to 100% for type 2, and 83.7% to 98.0% for type 3. SC rates 28 days after one dose were 32.7%, 36.7%, and 46.9% for types 1, 2, and 3, respectively. ((See Appendix A).

Median titers and GMTs of type-specific poliovirus-neutralizing antibodies are depicted in Table 3.

## 4. Discussion

The results from this study confirmed the safety and immunogenicity of Sabin tOPV in vaccinated children and vaccine-naïve infants, and these results could potentially contribute to the evaluation of novel oral polio vaccines in the future.

In Group 1, the SP rates were already high at baseline (>80%), as all subjects had received at least 1 prior polio vaccination, and 90% had received at least 3. Considering those high rates, SC rates were expected to be low; nevertheless, at 28 days after the single dose of tOPV, the SP rates increased to more than 94% for the three serotypes. In group 2, after three doses of tOPV, seroprotective levels were above 93% for the three serotypes, and SC rates ranged between 88% and 98% for types 1, 2, and 3 (PP population). These results showed that tOPV was immunogenic in both groups, children previously vaccinated and infants who had never received such vaccine. An interesting finding was the differences between the median titers and SP/SC rates at day 84 in group 2, where SC/SP rates were higher for type 3 than for type 1, but median titers were higher for type 1 as compared to type 3. A study in China that assessed comparisons between different polio vaccination schedules reported a similar result in tOPV recipients [19].

The vaccine was well tolerated in both groups, with no safety concerns observed during the study, and no AEs led to permanent discontinuation or death. For Group 1, there were no SARs, IMRs, or severe ARs. For Group 2, there were no SARs or IMRs, and although there were two severe ARs (vomiting after dose 1 and crying abnormal after dose 3), they occurred in a single subject and were the only ARs in the group. Both severe ARs were considered consistent with a causal association to the study vaccine. As expected, vomiting was the most frequent solicited adverse event following vaccination in infants. 

An expert review of the immunogenicity of tOPV in 15 developing countries reported significant variation in the detectable antibodies after three doses. The average of 73%, 90%, and 70% of SP for types 1, 2, and 3 were described, respectively, especially with schedules starting at 6 weeks of age and 1-month intervals between doses. Some hypotheses try to explain the differences, including concurrent infections with other enteroviruses, parasitic infections, colostrum secretory IgA, and differences in levels of maternal antibodies between populations [20].

There are no recent publications on the safety of tOPV regarding solicited and unsolicited AEs, SAEs, and IMEs in the age group reported. In a study conducted in India in 2012 that compared tOPV with IPV, 30 SAEs were reported, including one death, in 372 infants recruited for a booster dose. There were no significant differences in the SAE counts between the arms (11/186 in IPV and 9/186 in tOPV, *p*= 0.65), as well as in the vaccination-related AE counts (3/186 vs. 1/186, p = 0.62) [21]. As in our study, most of the events were hospitalizations due to respiratory illness and were not causally associated with the immunization.

As part of the eradication strategy, novel OPV-containing types 1 and 3 polio vaccines are being developed as monovalent and trivalent formulations with nOPV2 as they are expected to be genetically more stable than the Sabin OPVs [22,23].

This study provides a valuable reference point for future studies with novel OPVs, especially novel tOPV, given that Sabin type 2-containing vaccines can no longer be used in clinical trial settings for direct comparison due to containment-related recommendations [24].

## 5. Conclusions

In conclusion, tOPV was well tolerated and immunogenic in pre-vaccinated children and vaccine-naïve infants. One dose in pre-vaccinated children and the three-dose series in vaccine-naïve infants resulted in SP rates ≥90% for all poliovirus serotypes in both groups. These results suggest that tOPV is safe and immunogenic and may be used as a valid historical comparison for future studies investigating novel oral polio vaccines in similar age groups with comparable study designs. 

Potential data generated from new clinical studies with trivalent OPV formulations compared with the data reported here could enable the evaluation by regulatory authorities of new vaccines to be used in outbreaks and, hence, help interrupt poliovirus transmission.

## Figures and Tables

**Figure 1 vaccines-12-00953-f001:**
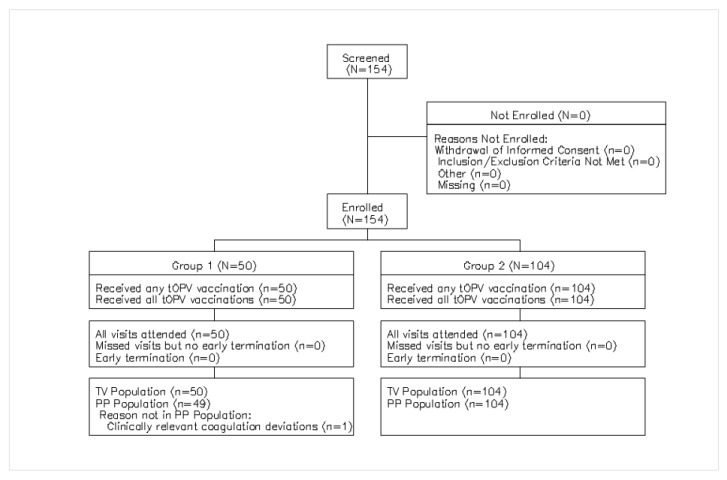
Population Disposition.

**Table 1 vaccines-12-00953-t001:** Summary demographic characteristics by group—Total Vaccinated Population.

Characteristic			Children (Group 1)	Infants (Group 2)
			(N = 50)	(N = 104)
Gender	Male	n (%)	27 (54.0)	48 (46.2)
	Female	n (%)	23 (46.0)	56 (53.8)
Age in weeks		Mean (SD)	148.3 (59.62)	5.8 (0.80)
		Median (Q1, Q3)	153.0 (101.0, 193.0)	6.0 (5.0, 6.0)
		Range (min, max)	(53, 257)	(5, 8)
Ethnicity	Hispanic	n (%)	50 (100)	104 (100)
Body weight [kg]		N	50	104
		Mean (SD)	14.1 (3.02)	4.9 (0.51)
		Median (Q1, Q3)	14.2 (12.2, 16.4)	4.9 (4.5, 5.3)
		Range (min, max)	(8.1, 20.0)	(3.7, 6.3)

Q1, upper limit of 1st quartile; Q2, lower limit of 3rd quartile; n, number of subjects with the indicated characteristic; N, number of subjects in group; SD, standard deviation.

**Table 2 vaccines-12-00953-t002:** Incidence and events of solicited AEs within 7 days of vaccination—Total Vaccinated Population.

Solicited AE	Causally Associated ^1^	Severity ^2,3^		Group 1 Days 0–7 (N = 50)	Group 2 Days 0–7 (N = 104)	Group 2 Days 28–35 (N = 104)	Group 2 Days 56–63 (N = 104)	Group 2 Total (N = 104)
All	Total	Any	n (%) m	7 (14.0) 9	27 (26.0) 43	18 (17.3) 29	7 (6.7) 15	39 (37.5) 87
	Causally associated	Any	n (%) m	0 (0.0) 0	7 (6.7) 8	4 (3.8) 6	3 (2.9) 4	12 (11.5) 18
Loss of appetite	Total	Any	n (%) m	2 (4.0) 2	5 (4.8) 5	1 (1.0) 1	2 (1.9) 2	7 (6.7) 8
		Severe	n (%) m	0 (0.0) 0	1 (1.0) 1	0 (0.0) 0	0 (0.0) 0	1 (1.0) 1
	Causally associated	Any	n (%) m	0 (0.0) 0	0 (0.0) 0	0 (0.0) 0	0 (0.0) 0	0 (0.0) 0
		Severe	n (%) m	0 (0.0) 0	0 (0.0) 0	0 (0.0) 0	0 (0.0) 0	0 (0.0) 0
Abnormal Crying	Total	Any	n (%) m	1 (2.0) 1	7 (6.7) 7	6 (5.8) 8	5 (4.8) 5	15 (14.4) 20
		Severe	n (%) m	0 (0.0) 0	1 (1.0) 1	0 (0.0) 0	1 (1.0) 1	2 (1.9) 2
	Causally associated	Any	n (%) m	0 (0.0) 0	2 (1.9) 2	3 (2.9) 4	2 (1.9) 2	6 (5.8) 8
		Severe	n (%) m	0 (0.0) 0	0 (0.0) 0	0 (0.0) 0	1 (1.0) 1	1 (1.0) 1
Drowsiness	Total	Any	n (%) m	0 (0.0) 0	2 (1.9) 2	1 (1.0) 1	2 (1.9) 2	4 (3.8) 5
		Severe	n (%) m	0 (0.0) 0	0 (0.0) 0	0 (0.0) 0	0 (0.0) 0	0 (0.0) 0
	Causally associated	Any	n (%) m	0 (0.0) 0	0 (0.0) 0	0 (0.0) 0	1 (1.0) 1	1 (1.0) 1
		Severe	n (%) m	0 (0.0) 0	0 (0.0) 0	0 (0.0) 0	0 (0.0) 0	0 (0.0) 0
Fever	Total	Any	n (%) m	5 (10.0) 5	5 (4.8) 6	8 (7.7) 9	2 (1.9) 2	13 (12.5) 17
		Severe	n (%) m	0 (0.0) 0	0 (0.0) 0	0 (0.0) 0	0 (0.0) 0	0 (0.0) 0
	Causally associated	Any	n (%) m	0 (0.0) 0	1 (1.0) 1	1 (1.0) 1	0 (0.0) 0	2 (1.9) 2
		Severe	n (%) m	0 (0.0) 0	0 (0.0) 0	0 (0.0) 0	0 (0.0) 0	0 (0.0) 0
Irritability	Total	Any	n (%) m	0 (0.0) 0	6 (5.8) 6	3 (2.9) 3	3 (2.9) 3	9 (8.7) 12
		Severe	n (%) m	0 (0.0) 0	2 (1.9) 2	0 (0.0) 0	0 (0.0) 0	2 (1.9) 2
	Causally associated	Any	n (%) m	0 (0.0) 0	1 (1.0) 1	1 (1.0) 1	0 (0.0) 0	2 (1.9) 2
		Severe	n (%) m	0 (0.0) 0	0 (0.0) 0	0 (0.0) 0	0 (0.0) 0	0 (0.0) 0
Vomiting	Total	Any	n (%) m	1 (2.0) 1	16 (15.4) 17	5 (4.8) 7	1 (1.0) 1	17 (16.3) 25
		Severe	n (%) m	0 (0.0) 0	2 (1.9) 2	0 (0.0) 0	0 (0.0) 0	2 (1.9) 2
	Causally associated	Any	n (%) m	0 (0.0) 0	4 (3.8) 4	0 (0.0) 0	1 (1.0) 1	4 (3.8) 5
		Severe	n (%) m	0 (0.0) 0	1 (1.0) 1	0 (0.0) 0	0 (0.0) 0	1 (1.0) 1

Solicited AEs were collected for 7 days following each dose. n: Number of participants, %: Percentage of participants, m: Event counts, ^1^ “Causally associated” = “consistent with causal association to immunization”, ^2^ Each participant event is only counted once for a given severity level. ^3^ Percentages for severity subclasses are out of the total events for that causally associated category.

**Table 3 vaccines-12-00953-t003:** Median and geometric mean type-specific poliovirus-neutralizing antibody titers by study Group—Per-protocol Population.

Day	Type		Group 1	Group 2
			(N = 49)	(N = 104)
0	1	N	49	104
		Median (log2) (LCI, UCI)	8.2 (7.2, 9.8)	4.2 (3.2, 5.5)
		GMT (LCI, UCI)	200.9 (116.2, 343.8)	35.9 (24.7, 53.2)
	2	N	49	104
		Median (log2) (LCI, UCI)	7.8 (7.2, 8.8)	3.8 (3.2, 4.2)
		GMT (LCI, UCI)	251.3 (174.0, 360.4)	16.4 (13.2, 20.6)
	3	N	49	104
		Median (log2) (LCI, UCI)	6.5 (5.8, 7.5)	2.5 (2.5, 2.5)
		GMT (LCI, UCI)	77.5 (51.2, 116.8)	8.5 (7.3, 10.0)
28	1	N	49	0
		Median (log2) (LCI, UCI)	10.5 (9.17, 10.50)	NC (NC, NC)
		GMT (LCI, UCI)	445.5 (272.7, 710.6)	NC (NC, NC)
	2	N	49	0
		Median (log2) (LCI, UCI)	9.8 (9.5, 10.5)	NC (NC, NC)
		GMT (LCI, UCI)	681.2 (511.1, 882.9)	NC (NC, NC)
	3	N	49	0
		Median (log2) (LCI, UCI)	8.5 (7.8, 9.5)	NC (NC, NC)
		GMT (LCI, UCI)	306.3 (201.7, 452.0)	NC (NC, NC)
84	1	N	0	104
		Median (log2) (LCI, UCI)	NC (NC, NC)	10.5 (10.5, 10.5)
		GMT (LCI, UCI)	NC (NC, NC)	650.8 (473.7, 870.6)
	2	N	0	104
		Median (log2) (LCI, UCI)	NC (NC, NC)	10.2 (9.8, 10.2)
		GMT (LCI, UCI)	NC (NC, NC)	712.2 (594.4, 840.3)
	3	N	0	104
		Median (log2) (LCI, UCI)	NC (NC, NC)	8.8 (8.5, 9.5)
		GMT (LCI, UCI)	NC (NC, NC)	352.1 (263.9, 462.9)

LCI, lower 95% confidence interval; NC, not calculated; UCI, upper 95% confidence interval. 95% confidence interval for median and GMT were obtained using the percentile bootstrap method. GMT was computed by using lower and upper quantitation limits as observed values.

## Data Availability

The authors will make the raw data supporting this article’s conclusions available upon request.

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
