# Peer review of "Safety and Immunogenicity of Trivalent Oral Polio Vaccine in Vaccinated Children and Vaccine-Naïve Infants: A Phase 4 Study"

_vaccines, 2024, doi:10.3390/vaccines12090953_

Round 1

Reviewer 1 Report

Comments and Suggestions for Authors

Overall Comments:

This is a very important study and provides needed data for the continued clinical development of novel OPV formulations. It will serve as necessary control data for novel trivalent formulations as tOPV (nor monovalent PV2) are currently able to be used in clinical trial designs. It is well written and concise and provides clear immunogenicity and safety endpoint data in both infant and child populations. As noted below more specifically, my only major general comment is related to the inclusion of mucosal immunogenicity data. Was fecal vaccine viral shedding measured? If so, please include that data. Otherwise, it does not appear that mucosal immunogenicity data is provided and thus reference to it should be removed from the introduction.

Specific Comments:

Introduction

Would suggest more strongly highlighting that following the switch in 2016, use of OPV2 is no longer allowable per global polio containment recommendations and that this study was performed prior to cessation of routine use of tOPV in the DR.

Lines 80-81 – Authors state the study was conducted to generate data on the safety and immunogenicity (humoral and intestinal) and safety of tOPV to provide historical control data however there is no intestinal immunogenicity data provided in the manuscript. If other investigations are planned to assess intestinal immunogenicity from blood samples or fecal shedding (preferably), please clarify and note if future publications are planned.

Line 124 - Clinical laboratory assessments (hematology, 124 chemistry, and coagulation analyses) were performed

-               What specific laboratory assays are performed and at what time points (all listed?)? Are these performed for safety or immunogenicity outcomes? If for safety, consider moving from immunogenicity to safety section. 

 Line 171 - This method-controlled type I error rate for the overall non-inferiority comparison at levels < 0.25"

-               Consider revising this sentence. It currently reads as a sentence fragment. Wondering if method-controlled should not be hyphenated?

Discussion

Line – 281-283: Even if speculative, additional comments or potential explanations for this interesting finding would be of interest to readers.  Is this explained by differences in PV 1 vs 3 baseline titers? Does it suggest serotype specific differences in antibody durability? Or serotype differences in vaccine virus replication and antibody responses?

Line 289 – Please briefly describe the two severe ARs in Group 2 (vomiting and irritability) and their causal relationship. A sentence regarding the expected observed frequency of vomiting following OPV in infants would be beneficial to readers.

Lines 291-296 – Reference #21 appears to be referenced incorrectly (it does not link to anything). Further description of this review would be helpful. Is it a meta analysis or expert review/opinion in a book. How many different location and what range of developing countries were included?

[Sutter, R. W.; Kew, O. M.; Cochi, S. L. Poliovirus Vaccine-Live. In Vaccines; Elsevier, 2008; pp 631–685. 410 https://doi.org/10.1016/B978-1-4160-3611-1.50030-1.]

Line 306 – Suggest “monovalent formulations” instead of “monovalent combinations”

Line 318- Suggest changing to to for:  “historical comparison for future studies”

 Lines 317-329. Run on sentence. Suggest breaking into two sentences and more clearly stating approval goals and what is meant by “field use” (distinction between outbreak response vs. routine use) for readers.

Comments on the Quality of English Language

English is overall very clear and well written. A few minor grammatical comments/suggestions are provided above for clarity.

Author Response

This is a very important study and provides needed data for the continued clinical development of novel OPV formulations. It will serve as necessary control data for novel trivalent formulations as tOPV (nor monovalent PV2) are currently able to be used in clinical trial designs. It is well written and concise and provides clear immunogenicity and safety endpoint data in both infant and child populations. As noted below more specifically, my only major general comment is related to the inclusion of mucosal immunogenicity data. Was fecal vaccine viral shedding measured? If so, please include that data. Otherwise, it does not appear that mucosal immunogenicity data is provided and thus reference to it should be removed from the introduction.

Response: The introduction used information regarding mucosal immunity to show how the tOPV vaccine has proven to raise intestinal immunity in the context of WPV2 and 3 eradication (ref #6). Originally, it was planned to assess the mucosal intestinal. However, it still needs to be performed as, currently, there is no data on novel trivalent OPVs in development with which to compare.

Specific Comments:

Introduction

Would suggest more strongly highlighting that following the switch in 2016, use of OPV2 is no longer allowable per global polio containment recommendations and that this study was performed prior to cessation of routine use of tOPV in the DR.

Response: Information regarding the use of OPV2 was added in lines 58-60, and a reference was added. Information about the study performed before the cessation of routine use of tOPV was added in line 84, before Materials and Methods, where the Dominican Republic is stated.

Lines 80-81 – Authors state the study was conducted to generate data on the safety and immunogenicity (humoral and intestinal) and safety of tOPV to provide historical control data however there is no intestinal immunogenicity data provided in the manuscript. If other investigations are planned to assess intestinal immunogenicity from blood samples or fecal shedding (preferably), please clarify and note if future publications are planned.

Response: information on intestinal immunogenicity was erased as it was not reported in the study.

Line 124 - Clinical laboratory assessments (hematology, 124 chemistry, and coagulation analyses) were performed

-               What specific laboratory assays are performed and at what time points (all listed?)? Are these performed for safety or immunogenicity outcomes? If for safety, consider moving from immunogenicity to safety section. 

Response: Information about clinical laboratory assessments was moved to safety, lines 159-160, and is now together with the time points.

 Line 171 - This method-controlled type I error rate for the overall non-inferiority comparison at levels < 0.25"

-               Consider revising this sentence. It currently reads as a sentence fragment. Wondering if method-controlled should not be hyphenated?

Response: We agree with the reviewer, that the term “method-controlled” should not be hyphenated, which makes the sentence easier to read.

Discussion

Line – 281-283: Even if speculative, additional comments or potential explanations for this interesting finding would be of interest to readers.  Is this explained by differences in PV 1 vs 3 baseline titers? Does it suggest serotype specific differences in antibody durability? Or serotype differences in vaccine virus replication and antibody responses?

Response: We share the interest of the reviewer, and probably this issue can be explored further in subsequent studies, but other than some historic references indicating lower replicative ability (or fitness) of the type 3 virus compared to the other types, and some impact from maternally-derived antibodies, there is no specific data from this study that could explain the finding. 

Line 289 – Please briefly describe the two severe ARs in Group 2 (vomiting and irritability) and their causal relationship. A sentence regarding the expected observed frequency of vomiting following OPV in infants would be beneficial to readers.

Response: information on the severe ARs and their causal relationship to the study vaccine was added in lines 306-308. Also, a sentence on the expected frequency of vomiting was added in lines 308-309.

Lines 291-296 – Reference #21 appears to be referenced incorrectly (it does not link to anything). Further description of this review would be helpful. Is it a meta analysis or expert review/opinion in a book. How many different location and what range of developing countries were included?

[Sutter, R. W.; Kew, O. M.; Cochi, S. L. Poliovirus Vaccine-Live. In Vaccines; Elsevier, 2008; pp 631–685. 410 https://doi.org/10.1016/B978-1-4160-3611-1.50030-1.]

Response: The reference was corrected (it appears now as reference #22), and the information request was added.

Line 306 – Suggest “monovalent formulations” instead of “monovalent combinations”

Response: We agree with the suggestion and was corrected in the sentence.

Line 318- Suggest changing to to for:  “historical comparison for future studies”

Response: thank you for the suggestion, it was changed as per the recommendation.

 Lines 317-329. Run on sentence. Suggest breaking into two sentences and more clearly stating approval goals and what is meant by “field use” (distinction between outbreak response vs. routine use) for readers.

Response: the suggestion was accepted, and the text was divided into two ideas. The new text could be found in lines 344 to 347.

Reviewer 2 Report

Comments and Suggestions for Authors

I found various small (?) errors 

- line 113 - definition of AR

- should clarify seropositive and seroprotective 

- is definition of "seroconversion" for group 1 standard - might it more appropriate be called just increase for those above positivity threshold at start ?  

- line 156 - wrongly called a trial

- line 201 says mean age group 1 was 12.3 months but given as 148 weeks in Table 1

- line 282 - confuses SP/SC and SC/SP rates 

  Overall, numbers are small and there is vast experience with tOPV. And a large literature dating back decades. I doubt the safety data in this paper contribute anything given the vast experience with this vaccine.

The data on infants are complicated by maternal antibodies. I did not understand how this taken into account, and wonder whether it is a standard method. It might have been helpful to show plots of before and after titres perhaps separately for different age groups       

How do these data compare with similar studies indifferent populations - I believe that OPV responses have generally been lower in poor/low hygiene/LMIC  populations than in wealthy high-hygiene populations. What are the consequent implications for interpreting these data ?   

Comments on the Quality of English Language

Could be more rigorous and critical 

Author Response

 I found various small (?) errors 

- line 113 - definition of AR

Response: the study protocol stated:

The co-primary objectives of the study was to assess the safety expressed as incidence of serious adverse reactions (SARs), severe adverse reactions (ARs) (grade 3 according to Common Terminology Criteria for Adverse Events [CTCAE] 4.03), and important medical reactions (IMRs) after one dose of SABIN tOPV in 1-5 year-old children and three doses of SABIN tOPV in 6 week-old infants, and the immunogenicity (seroprotection rates for all 3 serotypes) 28 days after three doses of SABIN tOPV in vaccine-naïve infants

In both cases, as AR stands for adverse reactions, the clinical protocol distinguished between serious adverse reactions with the SAR abbreviation, and for severe adverse reactions, severe AR was used.

- should clarify seropositive and seroprotective 

Response: the definition was aligned with the previous nOPV2 study, as our study was conceived with a similar study design to that study. “Safety and Immunogenicity of Two Novel Type 2 Oral Poliovirus Vaccine Candidates Compared with a Monovalent Type 2 Oral Poliovirus Vaccine in Children and Infants: Two Clinical Trials”, Ref #18 in the manuscript.

- is definition of "seroconversion" for group 1 standard - might it more appropriate be called just increase for those above positivity threshold at start ?  

Response: The definition of seroconversion is drawn from the relevant WHO TRS document #1045, which states:

Seroconversion for poliovirus antigen is defined as:

  • For subjects seronegative at the pre-vaccination time point, post-vaccination antibody titres of ≥8;
  • For subjects seroprotected at the pre-vaccination time point, a four-fold or greater rise in post-vaccination antibody titres. If the pre-vaccination titre is due to maternal antibodies, a four-fold rise above the expected titre of maternal antibodies based on the pre-vaccination titre declining with a half-life of 28 days indicates seroconversion, or post-vaccination antibody titres of ≥8, whichever is higher.

Reference: https://www.who.int/publications/i/item/9789240074484

- line 156 - wrongly called a trial

Response: Agree with the comment. The sentence was corrected.

- line 201 says mean age group 1 was 12.3 months but given as 148 weeks in Table 1

Response: The information was corrected to the accurate value of 148.3 weeks.

- line 282 - confuses SP/SC and SC/SP rates 

Response: The text corresponding to SP information was rearranged, and then the SC text was presented to be aligned with the previous paragraph. See lines 278 and 279.

Overall, numbers are small and there is vast experience with tOPV. And a large literature dating back decades. I doubt the safety data in this paper contribute anything given the vast experience with this vaccine.

Response: As noted in the manuscript, this data is expected to contribute as a reference point for future development of novel oral polio vaccines, especially the trivalent novel OPV formulation that is under development and expected to start clinical studies in 2024. Given the on-going public health emergency of international concern constituted by the spread of cVDPV2 and the massive use of novel OPV2 for outbreak response since 2021, it is anticipated that other novel OPV formulations, including trivalent nOPV will follow a similar development pathway as novel OPV2 and historic control data, such as from this study, will be important for future vaccines containing a type 2 poliovirus strain.

In addition, this data also adds to the existing literature of trivalent OPV data in the current times, where the polio epidemiology has evolved, with most geographies not having any passive exposure to wild polioviruses and minimal exposure to live OPVs based on current immunization schedules.

The data on infants are complicated by maternal antibodies. I did not understand how this taken into account, and wonder whether it is a standard method. It might have been helpful to show plots of before and after titres perhaps separately for different age groups     

Response: The seroconversion in infants is estimated relative to the projected level of maternally derived antibody at the post-vaccination serum sample dates. Briefly, the baseline sample and background knowledge of antibody decay kinetics is used to project the level of antibody derived from the maternal source that remains in the infant at post-vaccination visits (since it cannot be observed), and fold-rises (and therefore seroconversion rates) are computed as a function of this expected level of maternally derived antibody (and see response above related to the definition of seroconversion drawn from WHO guidance on the topic).

In supplements 2 and 3, we have attached the reverse cumulative distribution plots of poliovirus type-specific neutralizing antibody titers from both groups. We will include it in the main body manuscript at your discretion.

How do these data compare with similar studies indifferent populations - I believe that OPV responses have generally been lower in poor/low hygiene/LMIC  populations than in wealthy high-hygiene populations. What are the consequent implications for interpreting these data ?   

Response: The reviewer is absolutely right – heterogeneity of immune response for OPVs across different sub-populations (impacted by various factors, including pre-existing enterovirus infections, malnutrition, and other environmental factors) has been documented, and we expect this is still relevant. So there will be limitations of extrapolating these findings to all geographies.

Round 2

Reviewer 2 Report

Comments and Suggestions for Authors

No further comments 

Author Response

Thank you for your comments